# Emotional faces guide the eyes in the absence of awareness

Petra Vetter[1,2†]*, Stephanie Badde[1†], Elizabeth A Phelps[1,3], Marisa Carrasco[1]*

[1]Department of Psychology, Center for Neural Science, New York University, New York, United States; [2]Department of Psychology, Royal Holloway, University of London, Egham, United Kingdom; [3]Department of Psychology, Harvard University, Cambridge, United States

**Abstract** The ability to act quickly to a threat is a key skill for survival. Under awareness, threat-related emotional information, such as an angry or fearful face, has not only perceptual advantages but also guides rapid actions such as eye movements. Emotional information that is suppressed from awareness still confers perceptual and attentional benefits. However, it is unknown whether suppressed emotional information can directly guide actions, or whether emotional information has to enter awareness to do so. We suppressed emotional faces from awareness using continuous flash suppression and tracked eye gaze position. Under successful suppression, as indicated by objective and subjective measures, gaze moved towards fearful faces, but away from angry faces. Our findings reveal that: (1) threat-related emotional stimuli can guide eye movements in the absence of visual awareness; (2) threat-related emotional face information guides distinct oculomotor actions depending on the type of threat conveyed by the emotional expression.
DOI: https://doi.org/10.7554/eLife.43467.001

*For correspondence:
petra.vetter@rhul.ac.uk (PV);
marisa.carrasco@nyu.edu (MC)

†These authors contributed equally to this work

## Introduction

Detecting and reacting to potential threats in the environment is an essential skill for survival. Emotional information that indicates threat in the environment, such as a fearful or angry face, confers perceptual and attentional advantages compared to neutral information (*Carretié, 2014*; *Pourtois et al., 2013*; *Vuilleumier, 2005*): Emotional information enhances visual sensitivity (*Phelps et al., 2006*; *Fox et al., 2000*), potentiates effects of attention on visual sensitivity (*Ferneyhough et al., 2013*; *Bocanegra and Zeelenberg, 2009*; *Bocanegra and Zeelenberg, 2011a*; *Bocanegra and Zeelenberg, 2011b*; *Ohman et al., 2001*; *Lundqvist and Ohman, 2005*) and on appearance (*Barbot and Carrasco, 2018*), and gains preferential access to awareness (*Amting et al., 2010*; *Yang et al., 2007*; *Milders et al., 2006*; *Hedger et al., 2015*). The advantages of emotional information extend to actions. Eye (*Bannerman et al., 2009a*; *Bannerman et al., 2009b*; *Bannerman et al., 2010*; *Nummenmaa et al., 2009*) - or pointing (*Valk et al., 2015*) movements towards emotional –especially threat-related stimuli– are facilitated, whereas saccades away from these are delayed (*Belopolsky et al., 2011*; *Kissler and Keil, 2008*). Further, emotional information influences gaze trajectories, that is directly guides eye movements. Usually, people's eyes are attracted more towards emotional than neutral faces (*Mogg et al., 2007*; *Kret et al., 2013a*; *Kret et al., 2013b*), yet, sometimes gaze is repelled from threat-related –angry or fearful– faces (*Hunnius et al., 2011*; *Becker and Detweiler-Bedell, 2009*; *Schmidt et al., 2012*).

Remarkably, the recognition of a face's emotional expression does not require awareness of the face. (Some authors refer to 'awareness' and 'consciousness' interchangeably; in line with a recent review on the dissociations of perception and eye movements for neutral stimuli (*Spering and Carrasco, 2015*), we use the term awareness and operationally define it as the ability to make an explicit perceptual report). Cortically blind individuals are able to correctly 'guess' the emotions of faces

they cannot see (*de Gelder et al., 2005*; *de Gelder et al., 1999*; *Pegna et al., 2005*; *Tamietto et al., 2009*; *Bertini et al., 2013*; *Striemer et al., 2017*). In neurologically intact observers, emotional — especially threat-related — information they remain entirely unaware of, is prioritized over neutral information (*Hedger et al., 2016*), facilitates visual discrimination (*Bertini et al., 2017*), and alters subsequent perceptual (*Almeida et al., 2013*) and discrimination judgments differentially for the unseen emotion (*Zhan and de Gelder, 2018*). Emotional stimuli observers are unaware of also elicit physiological reactions, for example changes in skin conductance (*Esteves et al., 1994*), facial muscle activity and pupil dilation (*Tamietto et al., 2009*), and activate subcortical structures, for example the amygdala, pulvinar, basal ganglia and superior colliculus (*Tamietto and de Gelder, 2010*; *Jiang and He, 2006*; *Troiani et al., 2014*; *Troiani and Schultz, 2013*), as well as cortical structures (*Pessoa and Adolphs, 2010*). Furthermore, suppressed emotional stimuli influence oculomotor response times; when observers saccade towards color patches to report a mask's color, saccades initiation is slower departing from masked angry faces than from masked happy faces (*Terburg et al., 2012*). Despite this evidence indicating perceptual and attentional advantages, it is unknown whether suppressed emotional information can also directly guide actions or whether we have to become aware of the threat before we can act upon it.

Eye movements are the ideal testing ground for the relation between suppressed emotional information and actions, as recent evidence has shown that eye movements and visual awareness for neutral stimuli can be dissociated (*Spering and Carrasco, 2015*). Whether and how oculomotor actions are guided by unaware emotional information can be investigated by suppressing stimuli from awareness while measuring eye movements. The dissociation of eye movements and visual awareness reflects situations in which the direction of eye movements change in response to particular visual features, such as orientation or motion direction, even though observers are unable to report these features because they are not aware of them (*Spering and Carrasco, 2015*; *Spering and Carrasco, 2012*; *Spering et al., 2011*; *Rothkirch et al., 2012*; *Glasser and Tadin, 2014*; *Kuhn and Land, 2006*; *Simoncini et al., 2012*; *Tavassoli and Ringach, 2010*). In other words, suppressed neutral stimuli determine the trajectory of eye movements in the absence of awareness. Moreover, visual attention can modulate the processing of neutral stimuli and the eye movements they elicit even when observers are unaware of them (*Spering and Carrasco, 2012*).

Angry and fearful faces convey different types of threat, and elicit different reactions under awareness. Fearful faces indicate a potential indirect threat in the environment of the fearful person without indicating the source of the threat, suggesting more information is needed to appraise the situation (*Davis et al., 2011*). This ambiguity leads to increased amygdala activation (*Whalen et al., 2013*), heightens visual attention to the seen fearful stimulus location (*Phelps et al., 2006*; *Ferneyhough et al., 2013*) and attracts eye movements that enable detailed processing of the face and its environment (*Mogg et al., 2007*). Angry faces indicate direct threats (a person with an angry expression looking directly at the observer) leading to avoidance (*Schmidt et al., 2012*; *Marsh et al., 2005*) or freezing (*Roelofs et al., 2010*) responses. When looking at angry faces observers show a stronger startle reflex (*Springer et al., 2007*), stronger activation of the torso muscles (*Huis In 't Veld et al., 2014*), widening of the pupils (*Kret et al., 2013b*), and higher activation in brain areas associated with defense preparation (*Williams et al., 2005*; *Pichon et al., 2009*; *Kret et al., 2011*) than when looking at fearful faces. Threat processing is considered to evolve along two distinctive neuronal pathways, a fast sub-cortical and a slower cortical route (*LeDoux, 1998*). There is some evidence that fear information, relative to anger information, may be preferentially processed along the fast, subcortical route (*Luo et al., 2007*). This rich body of research suggests differential processing of fear and anger under awareness. Yet, when observers are presented with unmasked angry and fearful faces, both attract (*Mogg et al., 2007*; *Kret et al., 2013a*) or repel (*Hunnius et al., 2011*; *Becker and Detweiler-Bedell, 2009*) their attention and eye gaze equally strongly. Of note, there is only circumstantial evidence for distinct effects of emotional faces in the absence of awareness: Fearful faces have preferential access to awareness (*Yang et al., 2007*; *Milders et al., 2006*; *Troiani et al., 2014*; *Jusyte et al., 2015*), whereas angry faces remain suppressed longer (*Jusyte et al., 2015*; *Zhan et al., 2015*), both compared to neutral faces presented during continuous-flash suppression (CFS).

Our study had two goals: (1) To test whether threat-related emotional stimuli can guide eye movements in the absence of visual awareness. Were suppressed emotional faces to guide eye movements, their trajectory would depend on the location of the emotional stimuli. (2) To test

whether these oculomotor actions in the absence of awareness are distinct for different threat-related facial emotions. Were threat-related emotional information to guide eye-movements, the trajectories could be similar or differential, that is depend on the nature of the threat, direct — angry faces — or indirect — fearful faces.

We investigated these critical open questions by assessing whether different threat-related stimuli can direct oculomotor actions in the absence of awareness, as indicated by objective and subjective measures, and whether those actions are similar or differential by contrasting eye movements elicited by angry faces and fearful faces.

## Results

We rendered emotional faces unaware using continuous flash suppression (*Tsuchiya and Koch, 2005*; *Fang and He, 2005*). Low contrast face stimuli located in one of four quadrants of the visual display were presented to the non-dominant eye and a high contrast flickering mask was presented

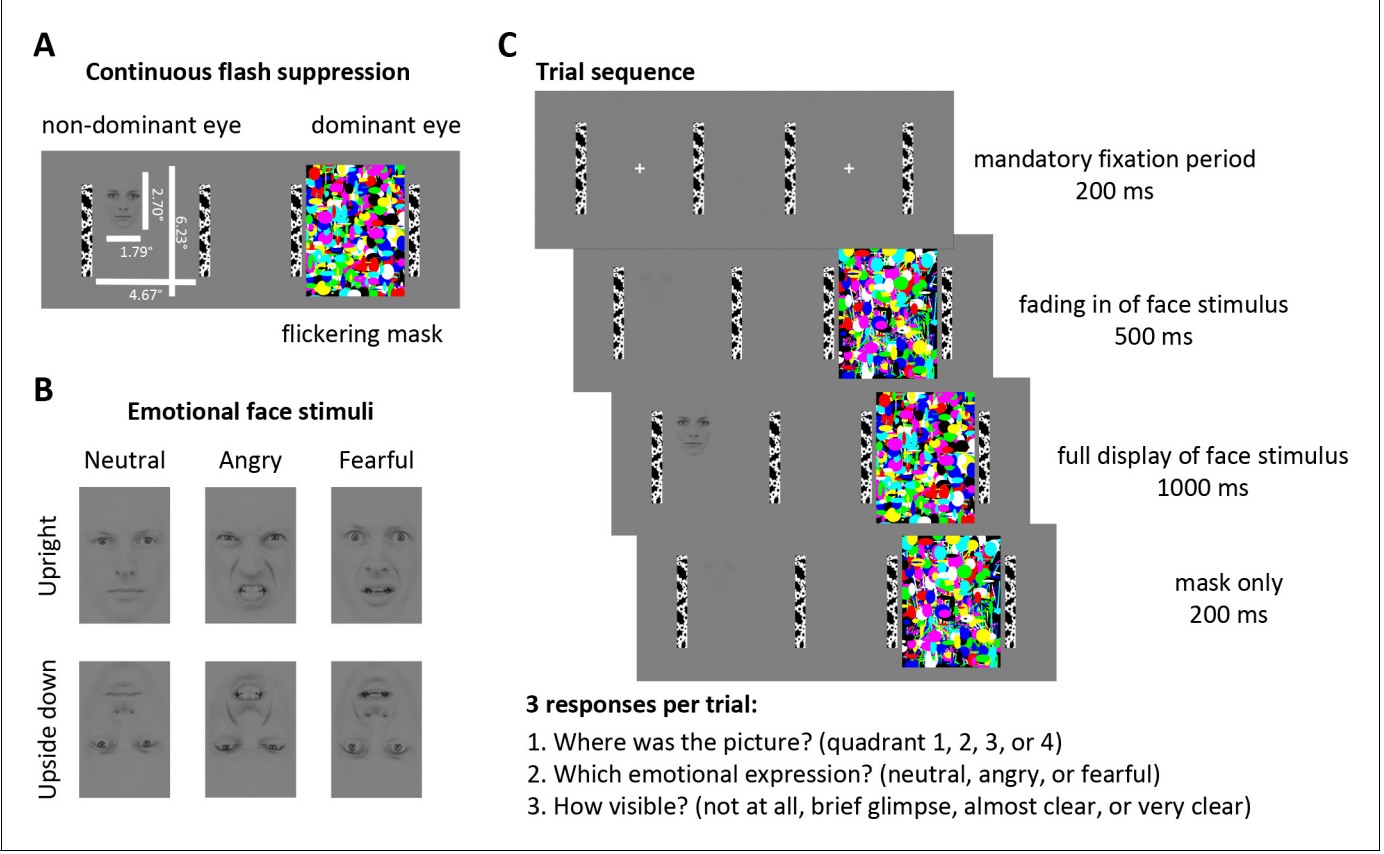

**Figure 1.** Stimuli and Experimental Design. (**A**) Continuous flash suppression was used to suppress low-contrast emotional face stimuli from awareness: when a high-contrast colored flickering mask is presented to the dominant eye (using a stereoscope) the viewer will not be aware of the picture presented to the non-dominant eye for several seconds. (**B**) Face stimuli with either a neutral, angry, or fearful expression (10 identities, five male, five female; see Materials and methods) were presented to the non-dominant eye. Faces were placed in any of the 4 quadrants of the stimulus field, either upright or upside down (to control for low-level visual features). (**C**) After a mandatory fixation period of 200 ms, the face stimulus was gradually faded in for 500 ms and fully presented to the non-dominant eye for 1000 ms while the flickering mask was continuously presented to the dominant eye. The mask was displayed for a further 200 ms to prevent aftereffects of the face stimulus. At the end of each trial, participants indicated the location of the face stimulus, its emotional expression, and its visibility by a button press.

DOI: https://doi.org/10.7554/eLife.43467.002

to the dominant eye (*Figure 1*). As such, the viewer is only aware of the flickering mask, but not of the stimulus presented to the non-dominant eye. In this way, continuous flash suppression suppresses stimuli for up to several seconds, enabling the measurement of eye movements in the absence of awareness (*Rothkirch et al., 2012*; *Madipakkam et al., 2016*).

To investigate the extent to which the different emotional facial expressions are processed and whether they affect eye movements differentially, we used two negative-valence expressions, *fearful* faces, indicating an indirect threat in the environment, and *angry* faces, indicating a direct threat from the depicted person. We also used *neutral* faces to control for the effects of face stimuli on eye movements that are unrelated to threat. To rule out the possibility that eye movements could have been simply triggered by low-level visual features (e.g., contrast differences between the eye and mouth regions), we also presented the same stimulus set upside down, as done in previous studies (*Phelps et al., 2006*; *Bocanegra and Zeelenberg, 2009*; *Barbot and Carrasco, 2018*) (*Figure 1*).

We determined full suppression of visual awareness based on subjective visibility ratings, whose validity we verified for each observer using objective measures. This verification is essential as subjective measures alone could reflect criterion rather than discriminability differences (*Hedger et al., 2016*; *Sterzer et al., 2014*). After each trial, participants (a) judged the location and emotional expression of the face stimulus and (b) rated its visibility (*Figure 2*). We analyzed only those trials in which subjective visibility was zero, *and* only data from participants' whose objective performance across trials with subjective zero visibility was at chance level for each of the two tasks, location and emotional expression, matching their subjective impression.

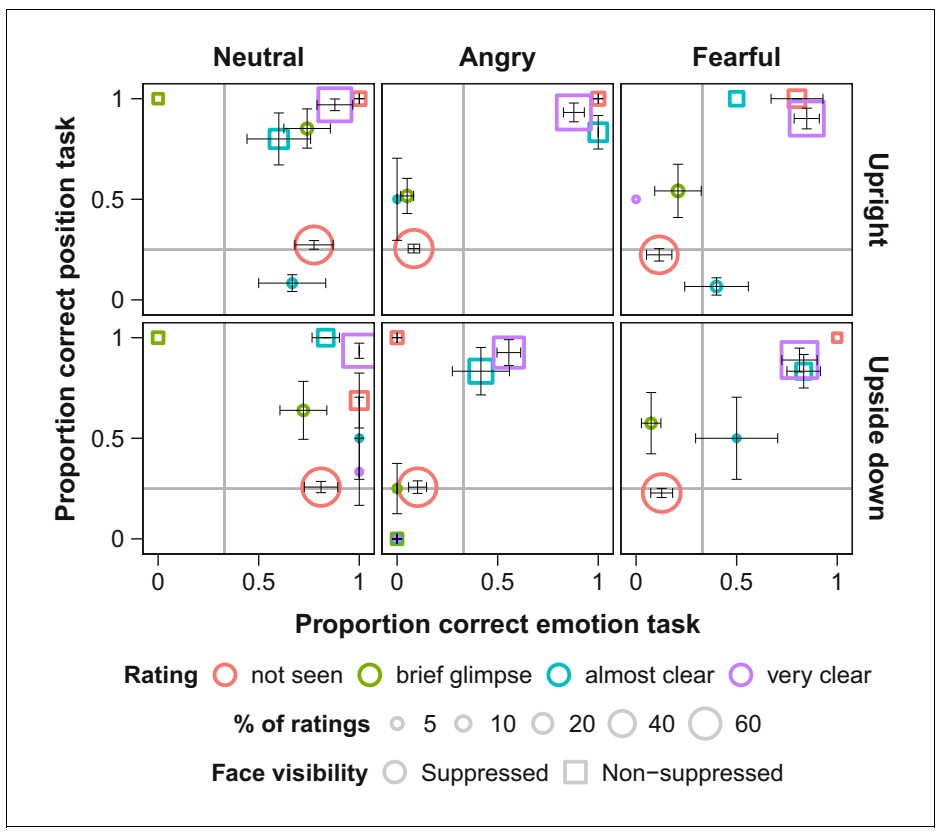

**Figure 2.** Objective and subjective measures of awareness of face stimuli presented under continuous flash suppression (CFS; circles) or on top of the flickering mask (squares). Proportion correct values for the position (y-axis) and emotion (x-axis) tasks are shown for each subjective rating of visibility (note that not all visibility ratings were selected in every condition). The area of the symbols corresponds to the average number of trials with the respective visibility rating per participant. Grey lines indicate chance level. Error bars show standard errors of the mean (SEM).

DOI: https://doi.org/10.7554/eLife.43467.003

Despite face stimuli being fully suppressed from awareness, participants moved their eyes towards fearful faces and away from angry faces, compared to neutral faces. *Figure 3* shows gaze position relative to the face stimulus as a function of time. Most locations within the display were further away from the face than the fixation point. Thus, the distance between gaze position and face increased for all faces when participants moved their eyes after an initial mandatory fixation period. At ~400 ms after full display was reached, distance of gaze position to fearful faces decreased compared to neutral faces, indicating an orienting of gaze towards fearful faces. In contrast, distance of gaze position to angry faces increased compared to the distance to neutral faces, indicating gaze aversion from angry faces. This pattern of results was not observed with upside-down presented faces. The absence of an effect for upside-down presented faces rules out the possibility that low-level visual features were responsible for the effect observed with upright presented emotional faces, a confound not controlled for in previous studies (*Bodenschatz et al., 2018*) (*Figure 3*).

To determine the preferentially viewed location for each emotion, we contrasted the spatial distribution of dwell times for upright faces with their upside-down counterparts; the latter provided a

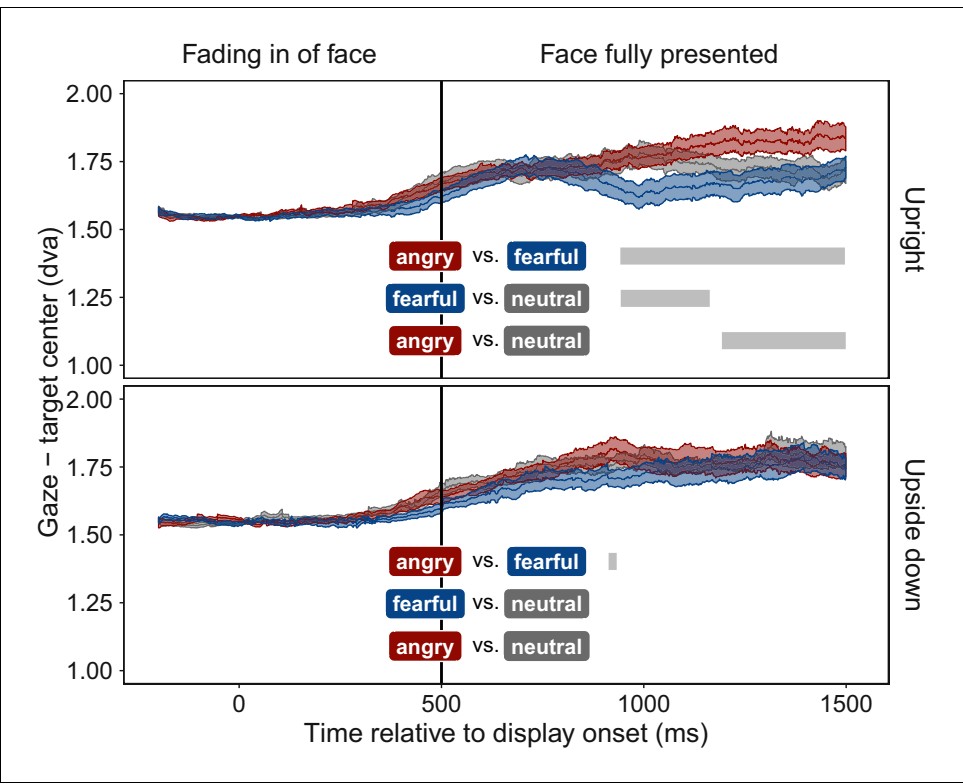

**Figure 3.** Time course of gaze distance from target. Mean distance of gaze position from the center of the face stimulus is plotted for all time points (±1 SEM shaded area) separately for upright (top panel) and upside down (bottom panel) presented face stimuli displaying fearful (blue), angry (red), or neutral (grey) emotions. Gaze data were included only for trials in which faces were fully suppressed from awareness (see *Figure 1*). After the fixation period, mean gaze distance increased in all conditions, as participants moved their eyes freely, and most areas of the display were located further away from the target than the initially-fixated center point. After ~400 ms of full stimulus display, mean gaze distance to upright fearful faces decreased, indicating an orienting towards fearful faces. In contrast, mean gaze distance to upright angry faces increased, indicating gaze aversion from an angry face. Grey bars represent significant clusters ($p < 0.001$) of adjacent time points (1 ms temporal resolution). At all time points within a cluster, a significant difference in distance between the two respective emotional expressions (colored textboxes) emerged ($p < 0.05$, corrected for multiple comparisons).
DOI: https://doi.org/10.7554/eLife.43467.004

The following figure supplement is available for figure 3:

**Figure supplement 1.** Time course of gaze distance from target overlaid with time course of average gaze distance from possible target locations for trials in which no target was presented (green).
DOI: https://doi.org/10.7554/eLife.43467.005

suitable baseline as they evoked no emotion-specific eye movements (*Figure 3*). This dwell time analysis (*Figure 4*) showed that, observers looked significantly longer towards the location at which an upright fearful face was displayed: dwell times at the target location increased by 57% compared to trials with upside-down presented fearful faces. In contrast, observers looked significantly longer towards an adjacent location when an upright compared to an upside-down angry face was displayed, demonstrating a consistent and lasting gaze aversion from an angry face. No effect emerged for neutral faces.

## Discussion

We found that in the absence of visual awareness, observers' eye movements were attracted towards fearful faces and repelled away from angry faces, compared to neutral faces. These novel results revealed that threat-related emotional faces can *guide oculomotor actions* without entering awareness. Our objective performance measures (*Figure 2*, red dots) validate the subjective visibility ratings, and together indicate that the emotional faces were suppressed from perceptual awareness. The absence of an effect with upside-down presented angry faces precludes confounds with low-level feature differences between faces of different emotional expressions.

Remarkably, the elicited eye movements were specific to the displayed facial emotion regarding the type of threat it conveyed. These distinct consequences on oculomotor actions demonstrate *qualitative processing* of the emotional expression, rather than mere detection of emotional stimuli in the absence of awareness. Previous studies on emotional face perception predominantly demonstrated preferential emergence of fearful faces into awareness (*Hedger et al., 2016*), which could simply indicate unspecific arousal caused by emotional content.

*How can emotional stimuli directly guide oculomotor actions in the absence of awareness?* For neutral stimuli, dissociations between visual awareness and eye movements, revealing more sensitive processing of information, have been related to the involvement of a fast subcortical retinocollicular pathway (*Spering and Carrasco, 2015*). Furthermore, processing of unaware emotional visual information has been associated with a subcortical pathway involving the amygdala, pulvinar, and superior colliculus (*Tamietto and de Gelder, 2010*), as well as structural connections between the

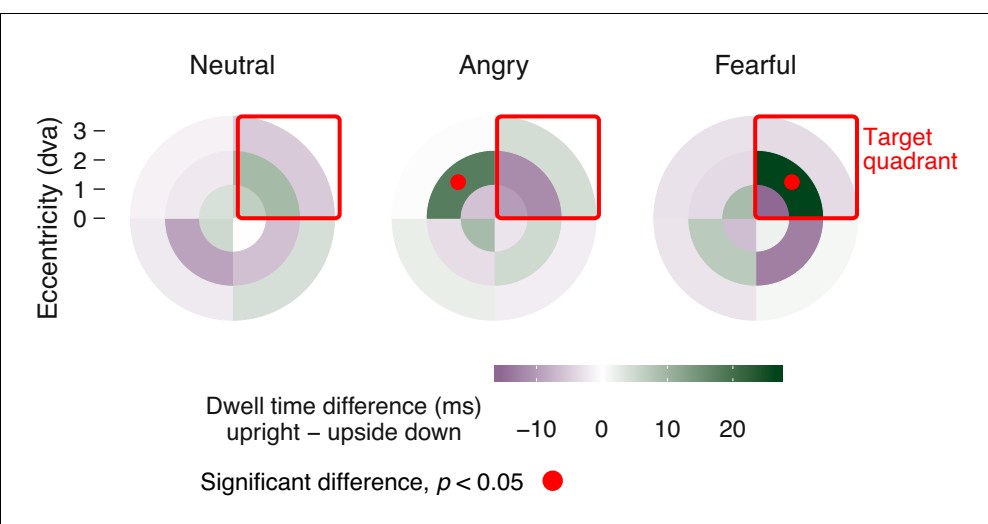

**Figure 4.** Spatial distribution of mean dwell time differences. Differences in dwell time between upright and upside down presented faces across the stimulus field, divided into four quadrants and three eccentricities. Data were aligned such that the target position is in the upper right quadrant (red square). Dwell time differences show an orienting towards the position of upright fearful faces (upright 62.82 ± 10.80 ms SEM, upside down 36.37 ± 6.45 ms; red dots: $p < 0.05$, corrected for multiple comparisons), and an aversion of gaze away from the position of upright angry faces (upright 51.57 ± 8.72 ms; upside down 33.62 ± 4.36 ms), both compared to upside-down presented faces of the same emotion.
DOI: https://doi.org/10.7554/eLife.43467.006

amygdala and cortical motor-related areas (*Grèzes et al., 2014*). The superior colliculus plays a crucial role in the control of voluntary and involuntary eye movements (*Bell and Munoz, 2008*). Thus, in the absence of awareness, orienting of eye movements towards and away from emotional faces could be mediated by these subcortical structures, in addition to the potential involvement of a cortical pathway (*Pessoa and Adolphs, 2010*). A possible mechanism could be that the amygdala assesses facial emotional expression, and the superior colliculus guides eye movements according to a fight, flight, or freeze response.

*Why did participants look towards fearful faces, but avert their gaze away from angry faces, in the absence of awareness?* As mentioned in the Introduction, under awareness these two threat-related emotions have differential perceptual consequences (*Kret et al., 2013b*; *Marsh et al., 2005*; *Springer et al., 2007*; *Huis In 't Veld et al., 2014*) and may be mediated by different neural pathways (*Luo et al., 2007*). In the absence of awareness, before this study, there had been only circumstantial evidence for differential processing of angry and fearful faces according to the time they take to break into awareness (*Yang et al., 2007*; *Milders et al., 2006*; *Troiani et al., 2014*; *Jusyte et al., 2015*; *Zhan et al., 2015*). With respect to eye movements, under awareness, both angry and fearful faces have shown to elicit similar eye movement responses, either attraction (*Mogg et al., 2007*; *Kret et al., 2013a*) or repulsion (*Hunnius et al., 2011*; *Becker and Detweiler-Bedell, 2009*). In the absence of awareness, gaze avoidance from masked angry faces had been demonstrated only indirectly with eyes starting instructed movements towards a response target more slowly in the presence of an angry than a happy face (*Terburg et al., 2011*). Instead, our results, directly measured with eye tracking, show uninstructed, sustained gaze avoidance to an angry face but sustained gaze attraction to a fearful face, while the faces are entirely suppressed from awareness. Fearful faces are more ambiguous and directing our eyes towards them may be an automatic response to try to gather more information (*Bannerman et al., 2009a*; *Mogg et al., 2007*; *Davis et al., 2011*). In contrast, angry faces pose a direct threat and directing our eyes away from them may be an automatic avoidance response (*Davis et al., 2011*).

Comparing our finding of differential eye movements in the absence of awareness with the previous findings of similar eye movements under awareness (*Mogg et al., 2007*; *Kret et al., 2013a*; *Hunnius et al., 2011*; *Becker and Detweiler-Bedell, 2009*) suggest that different factors may influence eye movements under aware and unaware conditions. Across studies, perceptual (*Lapate et al., 2016*), attentional (*Mogg et al., 1994*; *Fox, 1996*) and physiological (*Tamietto et al., 2015*) effects of emotional information either differ qualitatively or tend to be stronger in the absence than in the presence of awareness (*Diano et al., 2017*). Moreover, awareness modulates the time course (*Liddell et al., 2004*) and locus (*Tamietto et al., 2015*) of neural activations elicited by threat-related information, as well as functional connectivity between the amygdala and pre-frontal areas (*Amting et al., 2010*; *Lapate et al., 2016*; *Williams et al., 2006*). Two different mechanisms might cause threat-related emotions guiding actions differentially in the absence, but not in the presence of awareness: a) the perceived intensity of the threat might be stronger in the absence of awareness (*Lapate et al., 2016*; *Liddell et al., 2004*), which in turn may facilitate actions specifically tailored towards the nature of the threat, indirect or direct; b) cognitive mechanisms might suppress automatic actions upon threat-related information only under awareness (*Whalen et al., 2013*; *Mogg et al., 1994*), but not in its absence, as in our present findings. Several factors may help overwrite automatic eye movements towards fearful or away from angry faces, ranging from mere adherence to the task instructions (*Becker and Detweiler-Bedell, 2009*) to regulation of the social communication entailed in holding or averting gaze (*Terburg et al., 2011*; *Emery, 2000*; *Kendon, 1967*).

To conclude, our results provide the first evidence that emotional information humans are unaware of can differentially modulate their eye movements without entering awareness.

## Materials and methods

### Participants

Twelve participants (four females, mean age 24.3 years) were included in the final analysis. All participants took part in exchange for course credits, indicated normal or corrected-to-normal vision, and

signed an informed consent form. The experiment was conducted according to the guidelines of the Declaration of Helsinki and approved by the ethics committee of New York University.

## Stimuli and apparatus

We selected emotional face stimuli (neutral, angry, and fearful expressions, all looking straight ahead) from 10 different identities (five male, five female) from the Karolinska Directed Emotional Faces database (*Lundqvist et al., 1998*). The face images were cropped into oval shapes to remove hair, and their edges were smoothed to blend in with the grey background. Face stimuli were rendered low contrast, equated for overall luminance and contrast, and matched with the luminance of the background using the SHINE toolbox (*Willenbockel et al., 2010*) in MATLAB. Face images (1.79° x 2.70° visual angle) were presented either upright or upside down (*Figure 1B*) to control for low level visual features independent of emotional expression. The emotional expressions of upside down presented faces can elicit differential effects when these are presented for several seconds (*McKelvie, 1995*; *Calvo and Nummenmaa, 2008*; *Derntl et al., 2009*; *Narme et al., 2011*), but with shorter presentation times emotion-specific effects are limited to upright presented faces (*Phelps et al., 2006*; *Bocanegra and Zeelenberg, 2009*; *Barbot and Carrasco, 2018*). Faces were displayed in any of the 4 quadrants of the stimulus field (counterbalanced) to allow eye movements to be directed to different parts of the visual field. For continuous flash suppression, a set of colorful, high contrast flickering masks were created with randomly chosen shapes and colors, and each set of masks was created anew for each trial. Importantly, no mask image was ever exactly the same as another; therefore systematic features of the mask could not have directed eye movements. Mask images were displayed at a frequency of 28.3 Hz (*Kaunitz et al., 2014*). To help fuse the two images (*Figure 1*) while viewed through a stereoscope, black and white bars framed the stimulus field for each eye (4.67° x 6.23°). A mirror stereoscope (ScreenScope, ASC Scientific, Carlsbad, USA) was used to display different images to each eye. The flickering mask was always presented to the dominant eye (as determined with a hole-in-card test (*Miles, 1930*) at the start of the experiment for each participant) and the low contrast face stimulus was presented to the non-dominant eye to achieve best possible suppression. Stimuli were created and presented using MATLAB and the Cogent toolbox (Wellcome Department of Imaging Neuroscience, University College London) and displayed on a CRT monitor (IBM P260, 85 Hz, resolution 1280 × 1024). Participants were seated 114 cm from the monitor, placing their head on a chin rest and looking through the stereoscope. Eye movements were recorded with an infrared eyetracking system at 1000 Hz (EyeLink 1000, SR Research, Ottawa, Canada).

## Tasks

During mask presentation, participants were instructed to look for a face hidden behind the flickering mask. After each trial, they answered three questions by button press (maximum allowed response time was 2 s for each question). 1) Position task: Indicate the quadrant in which the picture was presented. 2) Expression task: Indicate the emotional expression of the face as angry, fearful, or neutral. 3) Visibility rating: Rate the visibility of the face stimulus as either 'not at all seen', 'brief glimpse', 'almost clear', or 'very clear '(perceptual awareness scale (*Sandberg et al., 2010*)). Two variants of numbering the quadrants and of assigning the keys to the three emotional expressions were counterbalanced across participants. The order of position and expression task was counterbalanced across participants, but the visibility rating was always administered last. The position and the expression tasks served as objective measures of awareness, that is determined whether participants were able to detect and identify the face stimulus above chance level. The visibility rating served as subjective measure of awareness. We only analyzed data from participants' whose performance in the localization and expression tasks was at chance when subjective visibility was rated zero (see Data Exclusion).

## Procedure

The experiment comprised 288 trials divided into three blocks to allow participants to rest and to recalibrate the eye tracker. In 240 trials, the face stimulus was presented to the non-dominant eye and the flickering mask to the dominant eye to suppress the face image from visual awareness. In 24 trials, the face image was displayed on top of the mask presented to the dominant eye, so that the

face stimulus was clearly visible (while overall visual stimulation intensity remained the same as during suppressed presentation). These non-suppressed trials were introduced as positive control, to ensure that participants paid attention and reported the image's position and emotion correctly when it was visible. In another 24 trials, only the mask was presented, with no face. These catch trials were included to measure baseline eye movement patterns and face position and emotion ratings without any face stimulus presentation to the non-dominant eye. Suppressed, non-suppressed, and catch trials were presented in random order, as were the face images' emotional expression, identity, orientation, and position.

Each trial started with a mandatory fixation period of 200 ms (*Figure 1C*), that is the trial was only initiated after participants maintained gaze at a central fixation cross for at least 200 ms. Then the flickering mask appeared to the dominant eye, and the face stimulus, presented to the non-dominant eye, was gradually faded in for 500 ms, to make the suppression more effective (*Tsuchiya et al., 2006*). The face stimulus was subsequently displayed to the non-dominant eye for 1000 ms before it disappeared. The mask was displayed for another 200 ms to prevent afterimages of the face stimulus.

## Data exclusion

The suppressive effects of continuous flash suppression vary inter- and intra-individually (*Hesselmann et al., 2016*). We employed subjective and objective measures to ensure that only trials in which the face stimulus was indeed suppressed from awareness were included in the analysis. First, only trials with a subjective visibility rating of zero ('not seen at all') were included in the analysis. Second, to validate these subjective judgments, participants' average accuracy in the emotion and in the position task was calculated separately for each level of subjective face visibility. Of special interest were trials in which participants reported to not have seen the stimulus (*Figure 2*, red circles). In the emotion task, participants were biased towards responding 'neutral' when they indicated that they had not seen the face (while at the same time they were at chance level for the position task; *Figure 2*). Yet, when averaging across trials all participants' proportion correct in the emotion task was close to chance level. In the position task, two participants performed clearly above chance level (78% and 67% correct position responses) even though they indicated to not have seen the stimulus. Data from these two participants were excluded from all further analyses. Two additional participants indicated to not have seen the face in only a low percentage of trials, thus, their data were too sparse for the analysis of gaze position over time (see below) and these participants were excluded from analyses of suppressed trials as well. For two participants we had only partial data due to problems with the hardware, these were excluded from the analysis. Additionally, less than 1% of the remaining trials were excluded from analysis because less than 25% of the recorded gaze positions were located inside the stimulus field (for example, because the participant repeatedly blinked during the trial).

## Data analysis

For the time series analysis (*Figure 3*), gaze position was evaluated as a function of time. To compensate for fluctuations in the eye-tracking device, single trial measurements were baseline corrected by subtracting the respective participants' average gaze coordinates during the fixation period from all measurements during the respective trial. To be able to analyze trials probing any of the four possible stimulus locations together, we calculated the Euclidian distance between gaze position and the center of the target quadrant for each time point in a trial. Mean distances were calculated per participant, time point, emotion, and face orientation. For each time point, these mean distances were compared between pairs of emotions, separately for each face orientation. For 87.82% of analyzed time points we had data from all participants, for the remaining 12.18% of timepoints data were missing from at least one of the participants. At $\alpha = 0.05$ the chances for false positive results, that is, significant test results without an underlying difference are at 5%. Each analyzed time segment comprised 1000 time points, resulting in a prediction of 50 false positive results per segment. To control for this alpha inflation, we employed permutation-based cluster mass tests (*Maris and Oostenveld, 2007*): In step 1, pairwise paired t-tests were calculated for each time point from 500 ms after trial onset onwards, that is each time point at which the face stimulus was fully displayed. Six t-tests for paired samples were conducted at each time point, comparing the average

distance between gaze and target within participants for all three possible pairings of the three emotional expressions, separately for upright and upside down presented faces. These t-tests were considered significant, if the absolute t-value was larger than 2. In step 2, clusters of adjacent time points with significant, equally directed differences between emotions were identified separately for each of the six comparisons (emotion pair x face orientation). That is, one temporal cluster comprised all sampled timepoints from start till end of the cluster and the t-tests conducted in step 1 had revealed a significant difference between the tested two emotions for all of these timepoints. For each emotion pair and face orientation the largest temporal cluster was determined based on the summed t-values of all time points within the cluster. These clusters were tested for significance by comparing these summed t-values to those of clusters derived from 1000 random permutations of the data. For each permutation, the emotion labels of a participant's gaze-target distances were swapped with a probability of 50%. Labels were swapped within but not across participants, and within one permutation the same permutated labels were used for all time points. Step 1 was repeated on the permuted data, and the largest cluster was extracted and compared to the corresponding cluster of the original data. A cluster was considered significant if clusters of the same or larger size occurred in less than 5% of the randomly permuted datasets. By doing so, we determined the probability that the original cluster of adjacent time points with significant differences between emotions could occur by chance.

Further, the distribution of dwell times across the stimulus field, that is, the cumulative time spent fixating a location on the screen, was analyzed (*Figure 4*). The analysis focused again on successfully suppressed trials, and – based on the results of the time series analysis – was restricted to gaze positions recorded at least 500 ms after full stimulus onset. Gaze position was transformed into polar coordinates and discretized by dividing the visual display into three equally spaced rings and four wedges. To allow averaging of trials across the four possible stimulus locations, the coordinates were rotated until the face was located in the upper-right quadrant. Average dwell times in each segment were compared between trials with upright versus upside down presented faces using paired t-tests, and this comparison was conducted separately for each emotion (step 1). Having established segments with significant differences in dwell time between trials with upright and upside down presented faces, we again used permutation cluster tests to avoid alpha inflation due to multiple comparisons (step 2). The clusters were now defined in space rather than time, that is, clusters were defined as adjacent segments with significant differences in the same direction. Minimal cluster size was one segment. Again, to evaluate the strength of a cluster, summed t-values were determined for each cluster within the original data and within 1000 permutations of the data. For the permutations, we randomly re-assigned the labels upright and upside down within each participant. The largest summed t-value of each permutation was extracted and the p-value of the largest cluster in the original data was determined based on the percentile of its summed t-value within this distribution. Thus, a spatial cluster (consisting of 1 or more adjacent segments of the stimulus space) was considered significant if the summed size of the effect within the cluster was not exceeded by more than 5% of clusters found in random permutations of the data.

Data and source files are available online (*Vetter et al., 2019*; copy archived at https://github.com/elifesciences-publications/EyeMovementsSuppressedEmotionalFaces).

## Acknowledgments

This study was supported by research fellowships from the Deutsche Forschungsgemeinschaft to PV (VE 739/1–1) and SB (BA5600/1-1) and by a grant from NIH-RO1-EY016200 to MC. We thank Jasmine Pan and Maura LaBrecque for help with data collection, members of the Carrasco Lab for discussions and comments on the manuscript, and Miriam Spering for comments on the manuscript.

## Additional information

### Competing interests

Marisa Carrasco: Reviewing editor, *eLife*. The other authors declare that no competing interests exist.

## Funding

| Funder | Grant reference number | Author |
|---|---|---|
| Deutsche Forschungsgemeinschaft | VE 739/1-1 | Petra Vetter |
| National Institutes of Health | NIH-RO1-EY016200 | Marisa Carrasco |
| Deutsche Forschungsgemeinschaft | BA 5600/1-1 | Stephanie Badde |

The funders had no role in study design, data collection and interpretation, or the decision to submit the work for publication.

## Author contributions

Petra Vetter, Conceptualization, Data curation, Software, Formal analysis, Funding acquisition, Investigation, Methodology, Writing—original draft, Project administration, Writing—review and editing; Stephanie Badde, Data curation, Software, Formal analysis, Funding acquisition, Validation, Investigation, Visualization, Methodology, Writing—original draft, Writing—review and editing; Elizabeth A Phelps, Conceptualization, Resources, Funding acquisition, Writing—review and editing; Marisa Carrasco, Conceptualization, Resources, Supervision, Funding acquisition, Investigation, Methodology, Writing—original draft, Project administration, Writing—review and editing

## Author ORCIDs

Petra Vetter  https://orcid.org/0000-0001-6516-4637
Stephanie Badde  http://orcid.org/0000-0002-4005-5503
Marisa Carrasco  http://orcid.org/0000-0002-1002-9056

## Ethics

Human subjects: All participants took part in the experiment in exchange for course credits and signed an informed consent form. The experiment was conducted according to the guidelines of the Declaration of Helsinki and approved by the ethics committee of New York University (IRB# 13-9582).

## Decision letter and Author response

Decision letter https://doi.org/10.7554/eLife.43467.009
Author response https://doi.org/10.7554/eLife.43467.010

# Additional files

## Supplementary files

• Transparent reporting form
DOI: https://doi.org/10.7554/eLife.43467.007

## Data availability

Source data and all analyses are available on Github (https://github.com/StephBadde/EyeMovementsSuppressedEmotionalFaces; copy archived at https://github.com/elifesciences-publications/EyeMovementsSuppressedEmotionalFaces).

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
