## [Decision Letter]

Thank you for submitting your article "The eyes react to emotional faces without perceiving them" for consideration by *eLife*. Your article has been reviewed by three peer reviewers, including Melvyn Goodale as the Reviewing Editor and Reviewer #1, and the evaluation has been overseen by Richard Ivry as the Senior Editor. The following individual involved in review of your submission has agreed to reveal their identity: Beatrice de Gelder (Reviewer #2).

The reviewers have discussed the reviews with one another and the Reviewing Editor has drafted this decision to help you prepare a revised submission.

Summary:

This is an interesting paper. Although there is certainly a lot of evidence for unconscious processing of emotional expression, this is the first study is the first to show differences in the direction of eye movements as a function of the nature of masked emotional expressions. Nonconscious emotion perception first in de Gelder 1999 showing that individuals with lesions of primary visual cortex can detect facial expressions in their blind field – and most recently by Striemer et al., 2017, showing that a patient with lesions of V1 – plus the face processing regions in occipito-temporal cortex – can nevertheless discriminate a happy vs. fearful facial expression and a happy vs. angry expression (albeit unconsciously in a 2AFC task) [1]. But the authors have taken this one rather large step forward and shown that the nature of the response (look to or away from) the masked facial expression can take place in quite unconsciously in neurologically intact individuals. This is not mere detection or discrimination – but instead is a specific directional action to an unseen facial expression. The result is similar to that reported by Bannerman et al., 2009, for visible fearful expressions, but importantly the emotional expressions in the current study are masked from awareness by CFS.

Essential revisions:

1) The authors report: "When observers are presented with unmasked angry and fearful faces, both attract their attention and eye gaze equally strongly, suggesting that different factors drive eye movements to angry faces under aware and unaware conditions." This seems puzzling and deserves some explanation. If the eye movements towards and away for fearful and angry faces are thought to be adaptive, as the authors suggest, then it would seem likely that the such behaviour would also occur when participants actually perceive the faces. Is it possible that if participants are aware of the emotional expression on a face presented as a photograph, they could suppress the 'automatic' tendency to look towards or away from the face? In any case, the authors need to discuss (briefly) why this should be the case.

2) The theoretical contribution and the novelty could be better highlighted. Some directly relevant studies that have used eye movement based measures (Bannerman et al., 2009) or CFS (Zhan, 2015, 2018a, b, c) need to be discussed in brief. Two other points; first, different responses to fearful and angry faces have been reported in the literature and therefore the authors need to signal both in the title and in the Introduction, how the actions they measured differ from others. Second, the explanation for the differences in the direction of the eye movements they observed is highly speculative. A bit more explication for the observed directional differences in warranted.

3) The authors should also look at the methodological paper by Zhan et al., 2018, getting into details of the psychophysical parameters that yield or do not yield reliable CFS. What is the rationale of using a mask frequency of 28.3 Hz?

4) In Results section paragraph three, it says that the authors "…only analyzed trials in which subjective visibility was zero, and only data from participants whose overall performance at zero visibility was at chance level…". Could the authors please explain how this overall performance was calculated? Was this done per trial or for a group of trials? How were the responses to the two objective questions (location and expression) combined to determine chance level?

5) Figure 2: why do participants seem to be doing so poorly in the conditions of 'almost clear' and 'very clear' visibility. It seems that, even in these conditions of subjective clear visibility, participants still cannot identify the position of the faces (for upright faces), and are quite poor at identifying angry or afraid expressions (again for upright faces). Does this mean that the subjective ratings are a very poor measure of actual visibility? If so, is this a valid measure to use at all? Also, could the authors please add a scale so that the reader can interpret the size of the circles? Finally consider showing performance for all non-suppressed trials in this figure.

6) For the time series analysis (subsection “Data Analysis”), paired t-tests seem to have been computed at each time point, across all participants – is that correct? Also, what were the criteria to identify clusters of significant t-values? The authors say that "clusters of adjacent time points with significant, equally directed differences between emotions were identified" – did they have to be all adjacent, without a single interruption? What was the minimum number of time points? Finally, what does the p-value correspond to?

7) For the analysis of dwell times, the authors need again to explain the criteria used to establish a cluster. For this analysis, the authors explain that the p-value corresponds to the percentile of the cluster's summed t-value in the distribution of all summed t-values (for largest clusters) obtained after permutations. I believe that this means that we can make a judgment about the size of the obtained cluster (how unlikely is to find a summed t-value of that size) but not about its spatial distribution. Wouldn't be more informative to test the probability of each segment being included in a cluster?

---

## [Author Response]

Essential revisions:1) The authors report: "When observers are presented with unmasked angry and fearful faces, both attract their attention and eye gaze equally strongly, suggesting that different factors drive eye movements to angry faces under aware and unaware conditions." This seems puzzling and deserves some explanation. If the eye movements towards and away for fearful and angry faces are thought to be adaptive, as the authors suggest, then it would seem likely that the such behaviour would also occur when participants actually perceive the faces. Is it possible that if participants are aware of the emotional expression on a face presented as a photograph, they could suppress the 'automatic' tendency to look towards or away from the face?. In any case, the authors need to discuss (briefly) why this should be the case.

We have clarified the information that had led to the quoted sentence, and we have revised this sentence. We specify that awareness modulates perceptual, attentional, and physiological effects of emotional information, as well as the neuronal activity associated with processing of emotional information. We can only speculate on the role of awareness for the guidance of actions by emotional information, as we are the first to demonstrate such effects in the absence of awareness.

2) The theoretical contribution and the novelty could be better highlighted. Some directly relevant studies that have used eye movement based measures (Bannerman et al., 2009) or CFS (Zhan 2015, 2018a, b,c) need to be discussed in brief. Two other points; first, different responses to fearful and angry faces have been reported in the literature and therefore the authors need to signal both in the title and in the Introduction, how the actions they measured differ from others. Second, the explanation for the differences in the direction of the eye movements they observed is highly speculative. A bit more explication for the observed directional differences in warranted.

We have highlighted the theoretical contribution and the novelty of our study and extended our discussion of studies investigating processing of emotional faces using eye movements or CFS. We extended our discussion of differential effects of angry and fearful faces and relate back to these differences when discussing our finding of differential eye movements to suppressed angry and fearful faces.

3) The authors should also look at the methodological paper by Zhan et al., 2018, getting into details of the psychophysical parameters that yield or do not yield reliable CFS. What is the rationale of using a mask frequency of 28.3 Hz?

We based the frequency mask on the first study testing a wide range of mask frequencies, which found CFS to be most effective at the mask frequency we used (Kaunitz et al., 2014). This is consistent with our observation that CFS was effective in suppressing the face stimuli from awareness. We note that the Zhan and colleagues (2018) paper had not been published when we conducted our experiment. They report that CFS is most effective with lower mask frequencies, especially 4 Hz and its harmonics (differences of 2-3 seen trials on average with 360 trials presented in total). In future studies, we will take this more recent finding into account.

4) In Results section paragraph three, it says that the authors "…only analyzed trials in which subjective visibility was zero, and only data from participants whose overall performance at zero visibility was at chance level…". Could the authors please explain how this overall performance was calculated? Was this done per trial or for a group of trials? How were the responses to the two objective questions (location and expression) combined to determine chance level?

To determine whether a participant performed above chance level in trials with subjective zero visibility, we calculated the proportion correct across a group of trials and compared it to the proportion correct corresponding to chance responses. There is no possibility to evaluate chance level performance on a single trial level. We analyzed responses to the two questions (location and emotional expression) separately, because participants may have based their subjective visibility ratings on either feature. We calculated the proportion correct for each task across all trials rated ‘not seen’, independent of face position, expression, and orientation. For the position task, chance level was at 25%. The two excluded participants responded correctly in 78% resp. 67% of trials. In the emotion task, several participants exhibited a strong response bias towards ‘neutral’. However, all participants’ percent correct in this task was close to the predicted 33%. We clarified in the main text and added information in the Materials and methods section.

5) Figure 2: why do participants seem to be doing so poorly in the conditions of 'almost clear' and 'very clear' visibility. It seems that, even in these conditions of subjective clear visibility, participants still cannot identify the position of the faces (for upright faces), and are quite poor at identifying angry or afraid expressions (again for upright faces). Does this mean that the subjective ratings are a very poor measure of actual visibility? If so, is this a valid measure to use at all? Also, could the authors please add a scale so that the reader can interpret the size of the circles? Finally consider showing performance for all non-suppressed trials in this figure.

Please note that as indicated by the size of the circles, subjectivity ratings other than ‘not seen’ were very rare; the average accuracies are based on very few trials only and few participants contributed to each cell. Whereas this low frequency of responses indicates that it is hard to evaluate the corresponding accuracy level for visibility ratings other than ‘not seen’, it does confirm that faces were successfully suppressed. Moreover, performance for trials in which the face was presented on top of the mask clearly indicates that participants could identify the position and the emotional expression of the faces, when these were not suppressed. As suggested, we have added results from these trials and a scale indicating the proportion of trials each mark represents to Figure 2.

6) For the time series analysis (subsection “Data Analysis”), paired t-tests seem to have been computed at each time point, across all participants – is that correct? Also, what were the criteria to identify clusters of significant t-values? The authors say that "clusters of adjacent time points with significant, equally directed differences between emotions were identified" – did they have to be all adjacent, without a single interruption? What was the minimum number of time points? Finally, what does the p-value correspond to?

Yes, for the time series analysis paired t-tests have been computed at each time point. Six t-tests for paired samples were conducted at each time point, comparing the average distance between gaze and target within participants for all pairings of the three emotional expressions, separately for upright and upside down presented faces. These t-tests were considered significant, if the absolute t-value was larger than 2 (which is slightly more conservative than using *p* < 0.05). Time points with significant differences were grouped into temporal clusters of adjacent time points. Any interruption, i.e., time point without a significant difference between the two emotions at hand, split the cluster. The minimal number of time points that could form a cluster was 1. The p-value of a cluster corresponds to the probability that a temporal cluster of the same or larger size as the largest cluster in the original data could occur by chance. In other words, the p-value of a cluster indicates how likely it is that two emotions differed as strongly and for as many adjacent time points as in the original data, given the variability in our dependent variable. We extended our description of the analysis by including this information.

7) For the analysis of dwell times, the authors need again to explain the criteria used to establish a cluster. For this analysis, the authors explain that the p-value corresponds to the percentile of the cluster's summed t-value in the distribution of all summed t-values (for largest clusters) obtained after permutations. I believe that this means that we can make a judgment about the size of the obtained cluster (how unlikely is to find a summed t-value of that size) but not about its spatial distribution. Wouldn't be more informative to test the probability of each segment being included in a cluster?

Indeed, cluster permutation tests are not spatially specific, as they only evaluate the strength of the cluster. Thereby, the strength of a cluster depends on the size of the cluster and the size of the effect at all included points. Spatial information is contained in the identity of the cluster, and usually evaluated afterwards (which voxels/sensors belong to the cluster). The same holds for temporal clusters and this non-selectivity is only possible because by default cluster permutation tests evaluate only the strongest cluster (which could be problematic, if two strong clusters concurrently exist). In our data, competing clusters were not a problem. In fact, cluster sizes never exceeded one segment. We still used cluster permutation tests, as they safeguard our results against the risk of false positives associated with the large number of t-tests involved in our analyses. By comparing the strength of the difference in one segment of the original data against differences in (clusters of) segments occurring by chance across the whole stimulus field we are more strict in our test than by applying a spatially selective method. We have added this information in the Materials and methods section.